# The diagnostic test accuracy of telemedicine for detection of surgical site infection: A systematic review protocol

**Ross Lathan**[1]*, **Misha Sidapra**[1], **Marina Yiasemidou**[2], **Judith Long**[1], **Joshua Totty**[1], **George Smith**[1], **Ian Chetter**[1]

**1** Academic Vascular Surgical Unit, Hull Royal Infirmary, Hull, United Kingdom, **2** Bradford Teaching Hospitals, Bradford, United Kingdom

* ross.lathan2@nhs.net

**Data Availability Statement:** No datasets were generated or analysed during the current study. All relevant data from this study will be made available upon study completion.

## Abstract

Since the COVID-19 pandemic there has been a rapid uptake and utilisation of telemedicine in all aspects of healthcare. This presents a key opportunity in surgical site infection surveillance. Remote follow up methods have been used via telephone, with photographs and questionnaires for post-operative reviews with varying results. This review therefore aims to comprehensively synthesise available evidence for the diagnostic accuracy of all forms of SSI telemedicine monitoring. The protocol has been established as per both PRISMA-P (S1 Table) and the Cochrane handbook for reviews of diagnostic test accuracy. Medline, Embase, CENTRAL and CINAHL will be searched using a complete search strategy developed with librarian input, in addition to google scholar and hand searching. All study designs with patients over 18 and undergone a primarily closed surgical procedure will be eligible. Index tests will include all forms of telemedicine and a subgroup analysis performed for each of these. Comparative tests must include face to face review, and all reference standards will be included again for sub-group analyses. Search results will be screened by two investigators independently with a third providing consensus review on disagreements. Methodological quality will be assessed using the QUADAS-2 tool, first validated by two investigators as per the Cochrane handbook. Exploratory analysis will formulate summary receiver operating characteristic curves and forest plots with estimates of sensitivity and specificity of the included studies. Sources of heterogeneity will be identifying and investigated through further analysis. Potential benefits of telemedicine integration in surgical practice will reduce cost and travel time to patients in addition to avoiding wasted clinic appointments, important considerations in a peri-pandemic era. To avoid missed or further complications, there must be confidence in the ability to diagnose infection. This review will systematically determine whether telemedicine is accurate for surgical site infection diagnosis, which methods are well established and if further research is indicated.

**Funding:** The authors received no specific funding for this work.

**Competing interests:** The authors have declared that no competing interests exist.

# 1. Background

## 1.1. Target condition being diagnosed

This review primarily aims to identify surgical site infections (SSI). The Centre for Disease Control and Prevention (CDC) defines SSI as an infection within 30 days of an operation or up to 90 days if an implant is left in place and the infection is related to an operative procedure [1]. SSI are further classified as 'superficial incisional SSI', 'deep incisional SSI', and 'organ/space SSI'; further details of these definitions can be found in S1 File [2].

Surgical site infections complicate over 30% of operations, depending on the type of procedure [3, 4]. Up to 60% of these present after discharge and so accurate and timely diagnosis requires intensive follow-up and potentially significant travel distances on the patient's behalf [5]. For patients, SSI may have a significant impact on morbidity and mortality with subsequent time and cost implications [6]. Unsurprisingly, the burden of infection encompasses healthcare providers too, with a recent UK study showing an association of SSI with a 92% increase in length of stay and an adjusted episode cost of £3040 [7]. Treatment for SSI can range from an oral course of antibiotics, to the need for reintervention (drainage or debridement), prolonged inpatient readmission and the subsequent risk of further morbidity (i.e. thrombotic and ischaemic events).

## 1.2. Index and gold standard tests

The current 'gold standard' for diagnosing SSI is a 'face-to-face' review using the US CDC criteria [2]. However, other scoring systems or criteria may be used during a face-to-face review. The ASEPSIS score uses weighted, objectively measurable criteria to identify surgical wounds as satisfactory healing, impaired wound healing or infected (S1 Table) [8]. A newly developed utility, the Bluebelle wound healing questionnaire (WHQ) has been validated in general surgery as a patient or clinician reported outcome measure to identify surgical site infection [9].

Diagnosis of SSI using remote, digitally based contact between patients and clinicians (telemedicine) is being investigated as the primary index test in this review. In this review we are focussing on digital remote follow-up. There are many ways of implementing this in practice. These include the use of photographic images and/or video, either in real time or deferred, telephone review and instant messaging. There may be other novel methods not listed here that are identified during the review process.

## 1.3. Clinical pathway

Patients undergoing surgery may present with SSI at one of three typical time points.

If an infection does occur, the first potential route of diagnosis is prior to discharge and within the first week. Infections at this point will depend on the surgical procedure, underlying comorbidities, and age, as these factors will influence typical length of stay (LOS) and wound healing. They are likely to be picked up on ward rounds or through dressing changes with the nursing team then highlighting issues to the surgeon. Face to face diagnosis using CDC criteria in this instance is straightforward as the patient has not left the department. The patient may have to undergo further observation, dressing changes and antibiotic course. Rarely, further imaging or intervention may be required.

The second point is typically patient initiated, after discharge but prior to any planned follow up, within 30 days of operation. Infections at this stage may not reach the surgeon's knowledge as they are often managed by the patient's primary care physician. However, without specialist input some infections progress in severity. Patients with evidence of deep incisional or organ/space infections may be referred to secondary care by the primary care team, may

present to the emergency department or may contact the surgical team directly through an aftercare number. Only once seen can the gold standard assessment take place. Further imaging, microbiology culture and sensitivity testing are often implemented to ensure appropriate and specific management. Further surgical intervention may be required.

Finally, patients may not be contemporaneously identified as having SSI. Delayed or missed mild SSI diagnosis may present when the patient arrives in clinic for review (often after 30 days postoperatively, or if this is telephonic, it may be apparent through the history). Patients with more severe infection usually will have presented at the emergency department by this point, but if missed can impact on morbidity. Management will focus on ensuring no ongoing infection and alleviating further complications.

Index tests in this setting will likely be used as a comparative to face-to-face, gold standard review at planned review points. Potential implications of digital remote follow up are early and avoidance of missed diagnosis within the 30-day window for the CDC criteria.

## 1.4. Rationale

Widespread technological innovation and adoption have been exponential in the 21$^{st}$ century. Sophistication of the mobile phone now allows for instantaneous communication all over the planet. Users can even transfer image and video data in real time. In 2019, 88% of individuals in the UK were estimated to own a smartphone [10]. Naturally, the use of technology in healthcare has too progressed. Telemedicine is the remote diagnosis and treatment of patients by use of technology [11]. Coined in the 1970s, this concept has broadened with the advent of the smartphone and mobile data. Mobile health (mHealth) is a contemporary classification whereby healthcare is supported by the use of mobile devices [12]. The use of telemedicine has enabled patients in isolated centres access to specialist review through transfer of medical imaging, and teleconsultations are coming into practice [13]. In surgery, the process of postoperative care is changing with the introduction of telemedicine.

In 2020, the COVID-19 pandemic caused a countrywide lockdown in the UK, and many other countries all over the world. Elective operations were cancelled, expanding already lengthy waiting lists. This also posed a challenge for outpatient follow up, as patients requiring review would be at risk of COVID-19 in attending the hospital, and departments adapted to comply with social distancing regulations, limiting the number of people allowed in outpatient spaces at any one time.

Remote follow-up has the potential to reduce unnecessary clinic visits providing benefits for both patients and healthcare providers. The rationale of this review, therefore, is to synthesise the current available evidence for using telemedicine to diagnose or exclude SSI in the context of post-operative follow up.

## 1.5. Objectives

**1.5.1. Research question.** Primary: Is digital remote follow-up accurate for the diagnosis of surgical site infection?
Secondary:

- What methods are used to facilitate SSI diagnosis?

- What are the limitations of telemedicine in SSI diagnosis?

This systematic review aims to assess the diagnostic accuracy of using telemedicine to identify SSI post-operatively. It will also aim to identify which methods are currently in use for this and any limitations of telemedicine methods in SSI diagnosis.

**1.5.2. Objectives.**   Primary: To determine the diagnostic accuracy of digital remote follow-up for the diagnosis of surgical site infection

Secondary objectives

- To determine what methods of digital remote follow-up have been used

- Evaluate the accuracy of different digital remote follow-up methods

- To determine limitations of digital remote follow-up

## 2. Methods

### 2.1. Protocol development

This protocol and review have been developed using the Cochrane handbook for reviews of diagnostic test accuracy and the Preferred Reporting Items for Systematic Reviews and Meta-Analyses (PRISMA) statements [14, 15]. In addition, this protocol is reported in line with the PRISMA statement for review protocols (PRISMA-P) which is attached in S1 Table [15].

### 2.2. Criteria for considering studies for this review

**2.2.1. Types of studies.**   There will be no restrictions on inclusion based upon prospective or retrospective study designs. Study designs which result in measures of test accuracy will be included, including randomised and observational studies. There will be no limitations on study sample sizes, or quality to thoroughly synthesise the available literature, but this will be accounted for in a quality of evidence assessment.

All study types will be included, and a sub group analysis performed for direct, fully paired and randomised studies.

Narrative and systematic review articles, letters and opinion pieces will be excluded from this review, however the reference lists of review articles will be hand searched for completeness.

**2.2.2. Participants.**   All patients over 18, who have undergone a procedure involving surgical incisions and followed up in a postoperative pathway will be eligible. Studies involving children under the age of 18 years and written in language other than English will be excluded from the review.

The study setting will vary depending on whether the index or reference test is being examined. The index tests can be performed anywhere as they are remote, the reference will be in a secondary care setting, likely clinics.

The index and reference tests will be applied on clinical assessment and on suspicion of surgical wound infection, no prior tests are required.

**2.2.3. Index and comparative tests.**   The index test to be reviewed is digital remote follow-up. Telemedicine is the remote diagnosis or treatment of patients using communications technology, which encompasses the term digital remote follow-up and will be used as a synonym in this review. Specifically, this entails the use of photographs or video, either deferred or in real time, telephone review and/or instant messaging. All index test methods will be evaluated through subgroup analyses.

The comparative tests are face to face review by a healthcare professional with the patient to directly observe the wound and obtain a history.

**2.2.4. Target condition.**   Surgical site infection as defined by CDC is the target condition for this review; infection within 30 days of surgery or within 90 days if an implant is left in place. This is further discussed in section 1.1. Some studies may categorise this further into superficial, deep and organ/space SSI (more details in S1 File).

**2.2.5. Reference standards.** The same diagnostic criteria will be used for both tests. The gold standard for assessment of SSI is the US CDC criteria which clearly define indication of infection. Other possible methods are the ASEPSIS score with infection clearly define at a score of 21 or greater, and bluebelle WHQ. The latter has been suggested with an infection cut off score of 6–8. All methods of remote diagnosis will be extracted and a sub-group analysis performed for each type.

## 2.3. Search methods for identification of studies

**2.3.1. Electronic searches.** Medline, Embase, CENTRAL and CINAHL databases will be searched from inception to the current date. Additional resources will be identified through google scholar. The search strategy has been developed in conjunction with an information specialist. An example search strategy for Medline or Embase is provided below;

(exp (Telemedicine)/ OR (remote consultation).mp OR (teleconsultation).mp OR (teleconsultation).mp OR (mobile health).mp OR (telehealth).mp OR (ehealth).mp OR (mhealth).mp OR (e*health).mp OR (e health).mp OR (m*health).mp OR (m health).mp OR (telephon*).mp OR (photograph*).mp OR (video*).mp OR (mobile app*).mp)

AND

(exp (surgical wound infection)/ OR (surgical wound dehiscence).mp OR surgical site infection).mp OR (postoperative infection).mp OR (SSI).mp OR (wound infection).mp OR (surgical wound complication).mp OR (post-surgical infection).mp OR (post operative infection).mp OR (post-operative infection).mp)

**2.3.2. Searching other sources.** Additional searches will be conducted through hand-searching the reference lists of included articles and excluded review articles.

## 2.4. Data collection and analysis

**2.4.1. Selection of studies.** Search results will be deduplicated and uploaded to the specialised online review tool, Rayyan. Study titles and abstracts will be screened by two investigators, RL and MS, independently. Any disagreement will be resolved by consensus decision from a third investigator. Articles included at this stage will be retrieved for full text screening, again by two authors acting independently. Those included after full text screening will go on to data extraction.

**2.4.2. Data extraction and management.** Data extraction will be into a bespoke designed spreadsheet (Microsoft Excel), hosted remotely and updated in real-time. Information on study design, country of origin, participant age and gender, sample size, surgery performed, drop out rates, time to follow-up and type of remote follow-up (photograph / video / telephone / other) will be extracted. Details on infection rates for remote and face to face methods as well as sensitivity and specificity of diagnosis will also be extracted. If information is available on specifics of superficial, deep and organ/space SSI rates, these data will also be extracted.

**2.4.3. Assessment of methodological quality.** Methodological quality will be assessed using the QUADAS-2 tool (S2 File and Table 1 within shows example tabulation for assessment of methodological quality.) [16]. Due to the nature of varying study design, systematic reviews of diagnostic test accuracy are often prone to heterogeneous results. In 2003, the quality assessment of diagnostic studies (QUADAS) tool was developed. This has since been revised as QUADAS-2 and is recommended for use in such reviews by the Agency for Healthcare Research and Quality, Cochrane Collaboration, and the UK National Institute for Health and Clinical Excellence [16].

Two authors will assess each manuscript as per the QADAS-2 tool, with a third independent author providing consensus review when discrepancies occur. Further details on the

QUADAS-2 process can be found in S2 File, with an example of the data extraction process in Table 1.

**2.4.4. Statistical analysis and data synthesis.** SSI diagnosed using any diagnostic criteria as part of a face-to-face review will be the reference standard. The patient will be the unit of analysis. Forest plots and receiver operating characteristic (ROC) curves will be produced as part of the initial, exploratory analysis and used to display estimates of sensitivity and specificity of the included studies.

Summary measures of sensitivity and specificity will be produced using a bivariate model for meta-analysis, if there are sufficient studies.

*2.4.4.1 Receiver operator characteristic curves.* Summary receiver operator characteristic (sROC) curves will be plotted using each study included as a data point. Confidence regions will also be calculated. Plots will be produced using metaDTA [17].

*2.4.4.2 Sensitivity analysis.* Sensitivity analysis based upon risk of bias will be evaluated. Studies with a high or unclear risk of bias identified by the QADAS-2 tool will be excluded in a separate analysis.

*2.4.4.3 Assessment of heterogeneity.* Several sources of potential heterogenetiy have been identified, and their effects will be investigated through the use of subgroup analysis and metaregression. The diagnostic criteria used to diagnose SSI (CDC, ASEPSIS, Bluebelle WHQ or others) may influence the test accuracy as different tools have been found to have poor correlation [18]. Other potential sources include the type of index test (subgroup analysis of photograph methods is planned) and the reviewer (i.e surgeon vs community healthcare worker).

## 3. Discussion

Integration of telemedicine has multiplied in recent years, exponentially so in response to the COVID-19 pandemic. One prospective application is remote diagnosis of SSI. Consolidation of telemedicine in surgical practice has the potential to reduce cost to both patient and care provider, as well as improved time implications for both parties. This would also reduce unnecessary visits to hospital clinic with a healthy surgical wound, an important consideration during a pandemic. To avoid further complications however and allow for confidence in diagnosis, telemedicine must be accurate in the detection of SSI. Previous studies have shown erythema detection to be difficult in review of wound images compare to face to face, which may influence a diagnosis of infection [19]. This review aims to comprehensively examine the accuracy for all methods of remote diagnosis of SSI thereby enabling evidence-based decision making on remote reviews of post-operative patients.

## Supporting information

**S1 Table. PRISMA-P (Preferred Reporting Items for Systematic review and Meta-Analysis Protocols) 2015 checklist: Recommended items to address in a systematic review protocol.** The PRISMA-P 2015 checklist for items to address in a systematic review protocol.
(DOCX)

**S2 Table. ASEPSIS score.** The ASEPSIS scoring method. A score of $\geq 21$ equates to the presence of SSI.
(DOCX)

**S1 File. SSI classification.** SSI classification into superficial incisional, deep incisional, organ/space infection.
(DOCX)

**S2 File. QUADAS-2.** The QUADAS-2 tool for assessment of methodological quality of studies.
(DOCX)

## Acknowledgments

*Identification:* Protocol for systematic review of diagnostic test accuracy

 *Update:* Primary review

 *Registration and no.:* **Prospero ID—CRD42021290610**

## Author Contributions

**Conceptualization:** Ross Lathan, Marina Yiasemidou, Judith Long, Joshua Totty, George Smith, Ian Chetter.

**Data curation:** Ross Lathan, Misha Sidapra, Joshua Totty.

**Formal analysis:** Ross Lathan, Joshua Totty.

**Investigation:** Ross Lathan, Misha Sidapra, Judith Long, Joshua Totty.

**Methodology:** Ross Lathan, Misha Sidapra, Marina Yiasemidou, Judith Long, Joshua Totty, George Smith, Ian Chetter.

**Project administration:** Ross Lathan, Marina Yiasemidou, Judith Long, Joshua Totty.

**Resources:** Ross Lathan.

**Software:** Ross Lathan.

**Supervision:** Marina Yiasemidou, Judith Long, Joshua Totty, George Smith, Ian Chetter.

**Validation:** Ross Lathan.

**Visualization:** Ross Lathan.

**Writing – original draft:** Ross Lathan, Joshua Totty.

**Writing – review & editing:** Ross Lathan, Joshua Totty, George Smith, Ian Chetter.

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
