## [Decision Letter · Decision Letter 0]

27 Jun 2022

PONE-D-22-01855Telemedicine for Surgical Site Infection Diagnosis: A Systematic Review ProtocolPLOS ONE

Dear Dr. Lathan,

Thank you for submitting your manuscript to PLOS ONE. After careful consideration, we feel that it has merit but does not fully meet PLOS ONE’s publication criteria as it currently stands. Therefore, we invite you to submit a revised version of the manuscript that addresses the points raised during the review process.

The manuscript has been evaluated by one reviewer, and his comments are available below.

The reviewer has raised a number of concerns. He requests improvements to the reporting of methodological aspects of the study such as the inclusion of information on the search strategy and the Boolean terms.

Could you please carefully revise the manuscript to address all comments raised?

We look forward to receiving your revised manuscript.

Kind regards,

Lorena Verduci

Staff Editor

PLOS ONE

Journal Requirements:

No

No

4. Thank you for stating the following in the Acknowledgments/Funder Section of your manuscript: 

Academic Department of Vascular Surgery, Hull Royal Infirmary

The funder has not influenced the development of this protocol.

No

5. Please amend your manuscript to include your abstract after the title page.

6. We note you have included a table to which you do not refer in the text of your manuscript. Please ensure that you refer to Table 1 in your text; if accepted, production will need this reference to link the reader to the Table.

Reviewers' comments:

Reviewer's Responses to Questions

**Comments to the Author**

1. Does the manuscript provide a valid rationale for the proposed study, with clearly identified and justified research questions?

Reviewer #1: Yes

2. Is the protocol technically sound and planned in a manner that will lead to a meaningful outcome and allow testing the stated hypotheses?

Reviewer #1: Yes

3. Is the methodology feasible and described in sufficient detail to allow the work to be replicable?

Reviewer #1: Yes

4. Have the authors described where all data underlying the findings will be made available when the study is complete?

Reviewer #1: No

5. Is the manuscript presented in an intelligible fashion and written in standard English?

Reviewer #1: Yes

6. Review Comments to the Author

You may also provide optional suggestions and comments to authors that they might find helpful in planning their study.

Reviewer #1: A well written and thought-out protocol

Needs to have search strategy and Boolean terms detailed in the protocol

7. PLOS authors have the option to publish the peer review history of their article (what does this mean?). If published, this will include your full peer review and any attached files.

Reviewer #1: **Yes: **Vaikunthan Rajaratnam

---

## [Author Response · Author response to Decision Letter 0]

6 Jul 2022

Editor comments

Thank you the manuscript has been updated accordingly.

No

Thank you, I have included the statement; 'The authors received no specific funding for this work.' in the cover letter. 

No

Thank you, I have included the statement; 'The authors declared that no competing interests exist.' in the cover letter.

4. Thank you for stating the following in the Acknowledgments/Funder Section of your manuscript: 

Thank you, I have removed the funding related text and included the statement regarding funding as above in response #2.

5. Please amend your manuscript to include your abstract after the title page.

Thank you. The abstract is included after the title page, page 2. 

We note you have included a table to which you do not refer in the text of your manuscript. Please ensure that you refer to Table 1 in your text; if accepted, production will need this reference to link the reader to the Table.

Thank you, reference to the table has been made. 

Captions have been included and the supplementary material moved to a separate file - 'supplementary material - appendix'

Thank you, references are correct. 

Reviewer #1: A well written and thought-out protocol 

Needs to have search strategy and Boolean terms detailed in the protocol 

Thank you for your comments, I have moved the search strategy from the appendix to the methods section [Lines 224-235, p10-11].

---

## [Editor Report · Decision Letter 1]

16 Aug 2022

The diagnostic test accuracy of telemedicine for detection of surgical site infection: a systematic review protocol

PONE-D-22-01855R1

Dear Dr. Lathan,

We’re pleased to inform you that your manuscript has been judged scientifically suitable for publication and will be formally accepted for publication once it meets all outstanding technical requirements.

Kind regards,

Vaikunthan Rajaratnam

Guest Editor

PLOS ONE
---

## [Editor Report · Acceptance letter]

1 Sep 2022

PONE-D-22-01855R1 

The diagnostic test accuracy of telemedicine for detection of surgical site infection: a systematic review protocol 

Dear Dr. Lathan:

I'm pleased to inform you that your manuscript has been deemed suitable for publication in PLOS ONE. Congratulations! Your manuscript is now with our production department. 

Kind regards, 

on behalf of

Dr. Vaikunthan Rajaratnam 

Guest Editor

PLOS ONE